# Analysis of Tomographic Images of a Catastrophic Gas Embolism during Endoscopic Retrograde Cholangiopancreatography

**DOI:** 10.3390/diagnostics14131425

**Published:** 2024-07-03

**Authors:** Marta Frydrych, Marceli Łukaszewski, Kamil Nelke, Maciej Janeczek, Agata Małyszek, Jan Nienartowicz, Grzegorz Gogolewski, Maciej Dobrzyński

**Affiliations:** 1Sokolowski Specialist Hospital in Walbrzych, Sokolowski, 58-309 Walbrzych, Poland; frydrych.marta.k@gmail.com (M.F.); marceliluk@gmail.com (M.Ł.); 2Private Practice of Maxillo-Facial Surgery and Maxillo-Facial Surgery Ward, EMC Hospital, Pilczycka 144, 54-144 Wrocław, Poland; 3Health Department, Academy of Applied Sciences Angelus Silesius in Wałbrzych, Zamkowa 4, 58-300 Walbrzych, Poland; 4Department of Biostructure and Animal Physiology, Wrocław University of Environmental and Life Sciences, Kożuchowska 1, 51-631 Wrocław, Poland; maciej.janeczek@upwr.edu.pl (M.J.); agata.malyszek@upwr.edu.pl (A.M.); 5Private Practice of Maxillo-Facial Surgery, Romualda Mielczarskiego 1, 51-663 Wrocław, Poland; nienartowicz@gmail.com; 6Department of Emergency Medicine, Wrocław Medical University, Borowska 213, 50-556 Wrocław, Poland; grzegorz.gogolewski@umw.edu.pl; 7Department of Pediatric Dentistry and Preclinical Dentistry, Wrocław Medical University, Krakowska 26, 50-425 Wrocław, Poland; maciej.dobrzynski@umw.edu.pl

**Keywords:** ERCP, gas embolism, pulmonary embolism, SCA

## Abstract

Endoscopic retrograde cholangiopancreatography (ERCP) is a commonly performed minimally invasive procedure. Air embolism in a patient undergoing ERCP is relatively rare, accounting for approximately 2–3% of procedures performed, and a catastrophic air embolism is even rarer. Symptoms of air embolism can come from the cardiopulmonary and nervous system. It is important to remember this in the differential diagnosis of complications of ERCP, as early detection is crucial. In the case presented here, the diagnostic CT scan performed immediately after the incident brings awareness of how massive an air embolism can be. The CT results showed gas bubbles entering both the superior and inferior vena cava. The presence of air has been captured in the bile ducts, duodenum wall, heart, femoral veins and intracranially. Risk factors for this complication include previous biliary surgeries, the presence of prostheses and stents, cholangitis, liver tumors and anatomical anomalies such as hepatobiliary fistulas, as well as intrahepatic and extrahepatic anatomical leaks. As gas embolism is associated with serious health consequences, knowledge of the problem and adequate preparation may reduce the occurrence of the problem. Attention should be paid to basic and easily obtainable precautions when performing the procedure, such as the patient’s hemodynamic status, adequate hydration and positioning during the procedure.

We present the case of a 79-year-old female patient who underwent ERCP for choledocholithiasis. The patient had previously undergone ERCP after biliary stent placement. This time the procedure was terminated prematurely due to sudden cardiac arrest (PEA-type SCA). Rapidly initiated CPR was prolonged and continued after the patient was transferred to the ICU with a good return of efficient circulation. Once the cardiovascular system was stabilized and oxygen exchange was normal, a CT scan of the abdomen and pelvis (Figure 1, Figure 2 and Figure 3), an angio-CT scan of the chest (Figure 4), and a CT scan of the head (Figure 5) were performed to diagnose the cause of the SCA. The CT scan results revealed the etiology of the deterioration and visualized the etiopathogenesis of the severe complication of the procedure. The images of the catastrophic massive embolism show the mechanism of gas bubble transmission into the vascular system (Figure 1, Figure 2, Figure 3, Figure 4 and Figure 5).

The phenomenon of air embolism during endoscopic interventions is a serious medical problem. However, the exact incidence of embolic complications is unknown due to the fact that a certain proportion of embolic cases presents with a benign clinical picture [1]. An estimated 2.4% of all ERCP procedures are for embolic syndromes with severe clinical manifestations, including extreme fatal cases [2]. In contrast, it is likely that the majority of abortive embolic complications are sparsely symptomatic or have discrete clinical manifestations that are associated with invasive surgical procedures or an aesthetic management. The ERCP procedure, like most endoscopic procedures, uses CO_2_ gas insufflation to allow the visualization of the bile ducts and transmission of the endoscope through the gastrointestinal tract into the biliary tract.

Depending on the pathophysiology, air embolism can be divided into the following types: venous air embolism, as it reaches the pulmonary circulation; arterial air embolism, as it reaches the coronary or cerebral arteries; and paradoxical air embolism in which venous embolism reaches the systemic circulation. Figure 2 illustrates the onset of pathology caused by an ERCP complication: the presence of intramural CO_2_ gas in the duodenum. Subsequently, the gas is transmitted into the portal venous system through damaged duodenal vein radicles.

The pathway of CO_2_ gas transmission also occurs by the biliary tract; the gas then enters the vessels of the circulatory system through the injured endothelium. The pathomechanism and incidence of this complication are multifactorial. The type of endoscopes used and the working pressure of the gas used, the type of intervention, the presence of biliary-venous fistulas, the state of the mucosal barrier, previous endoscopic interventions or the presence of implanted foreign materials are all significant [3]. All these factors result in a breach of the endothelial and tissue barrier with the transmission of gas into the vascular system—the portal vein runoff or the hepatic venous system. Figure 1 shows bile ducts filled with gas. The transmission of CO_2_ into the superior vena cava system occurred via damage to the endothelial tissue barrier between the bile ducts and the hepatic veins.

The most common mechanism of air embolism involves gas bubbles entering the inferior vena cava and then the right heart cavities. The use of highly blood-soluble carbon dioxide aims to reduce the risk of this complication. When there is a massive accumulation of gas bubbles in the cavities of the right heart and pulmonary artery system, the course of the embolism is often detrimental. By performing a CT angiography of the thorax shortly after the SCA, we were able to capture CO_2_ gas in the right ventricle, as seen in Figure 4.

Cases of central embolism have been described in which gas bubbles enter the arterial system. This is favored by the presence of leaks in the heart as well as arteriovenous shunts in the pulmonary vascular system [4,5]. Embolic incidents with the direct entry of gas bubbles into the arterial system have also been described [1,2,3].

Due to the high solubility of carbon dioxide, the diagnostic imaging performed in a short interval after the SCA exceptionally illustrates the mechanism of gas bubble transmission into the vasculature of both vena cavae. We were able to visualize the presence of CO_2_ gas in the CT scan of the head in Figure 5 and in the distal section of the CT scan of the abdomen and pelvis which includes a section of the lower extremities in Figure 3. The explanation for such extensive gas transfer and its presence in the internal jugular vein, craniofacial tissues, sigmoid sinuses of the superior vena cava (Figure 5) and in the lumen of the femoral veins (Figure 3) may be explained by the CPR performed, and the possibility of very high pressures of the gas used, which may have been facilitated by the hypovolemic position and status of the patient.

In addition to the described mechanical disturbance of blood flow and the secondary disturbance in the blood supply to, e.g., myocardium or liver function, the presence of gas bubbles induces a specific type of inflammatory response. The activation of cytokines and platelet aggregation are manifested by sudden disturbances in the coagulation system [6,7]. Cellular toxicity also causes a rapid increase in kinase levels, liver enzymes and procalcitonin [8,9]. Considering the vascular abnormalities, characterized by the presence of gas bubbles in the vasculature, we observed a rapid increase in markers of infectious and enzymatic diseases. It is interesting to note the rapid increase and then equally rapid normalization of the parameters presented in Table 1.

Massive embolization by gas insufflated through the endoscope confirms the cases described in the literature, but such a massive example directly demonstrates the transmission of CO_2_ bubbles into the hepatic venous system and parallel to the portal venous system, followed by the displacement of gas bubbles into the tributaries of the superior and inferior vena cava, and even observed in the branches of the femoral veins. During the ERCP procedure, the patient suffered an extremely severe complication of massive pulmonary gas embolism, including SCA. The CT study was performed immediately after the CPR procedure, which was completed with the return of the patient’s own efficient circulation. Effective resuscitation and subsequent HBO therapy contributed to the therapeutic success and the patient survived without injury.

The aim of this article is to provide a unique opportunity to see the imaging studies depicting the path of post-ERCP air embolism. Adequate patient preparation and attention to predisposing factors are of great importance. Patient preparation for the procedure, including starvation and the preparation of the gastrointestinal tract, promotes dehydration. Prolonged duration of the procedure, hemodynamic compromise and advanced age should raise awareness of potential problems. Patient preparation should include the adequate filling of the vascular bed and limitation of the described exposures.

## Figures and Tables

**Figure 1 diagnostics-14-01425-f001:**
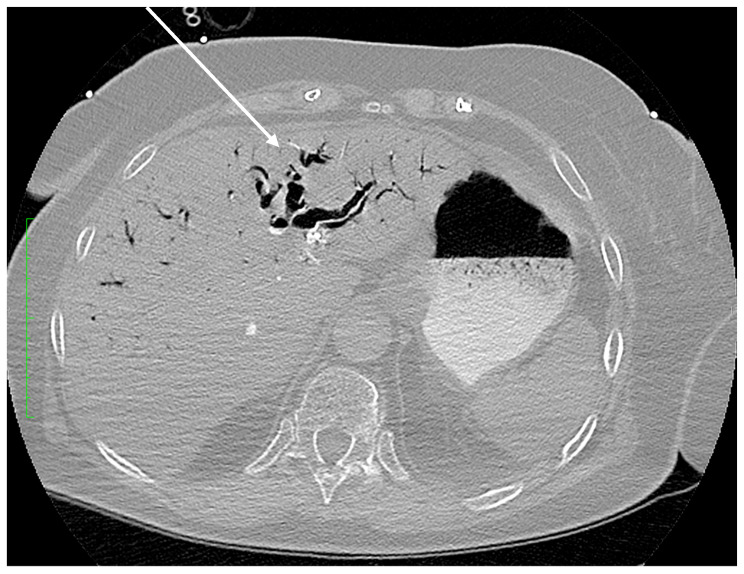
Axial plane of the abdominopelvic CT scan after intravenous contrast administration (lung window) at the level of the liver. Extensive intrahepatic gas is visible within the bile ducts (arrow).

**Figure 2 diagnostics-14-01425-f002:**
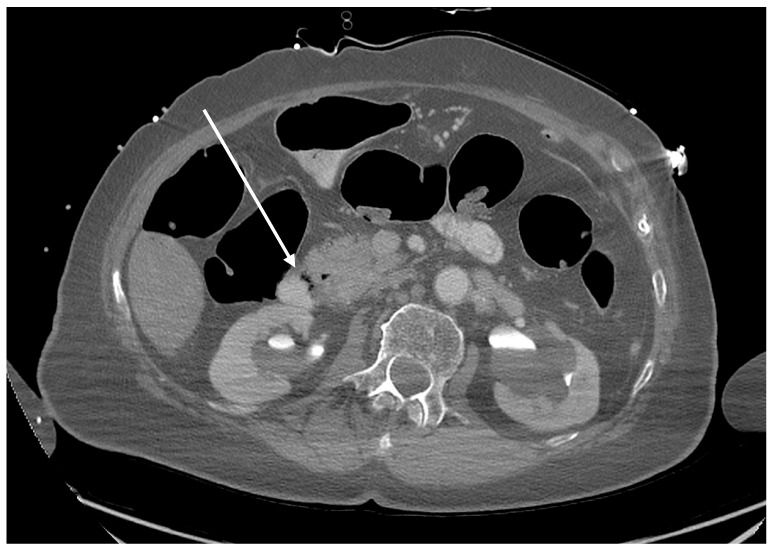
Axial plane of the abdominopelvic CT scan after intravenous contrast administration (delayed phase). Gas bubbles are detected in the duodenum wall (arrow).

**Figure 3 diagnostics-14-01425-f003:**
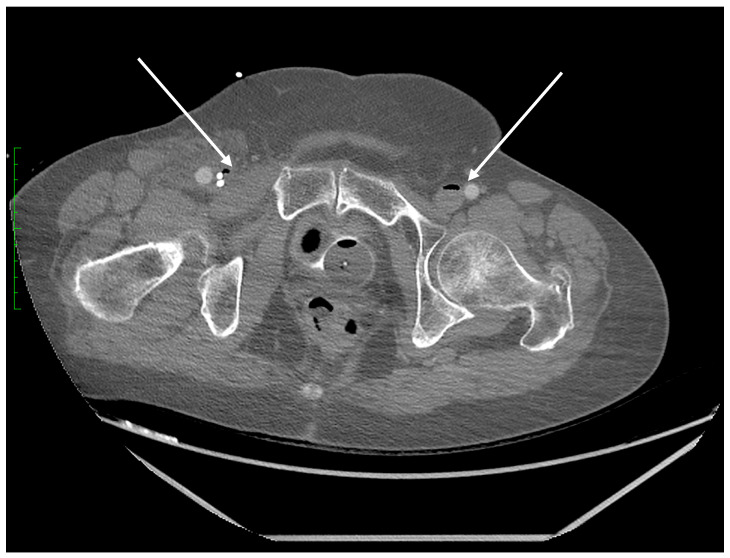
Axial plane of the abdominopelvic CT scan after intravenous contrast administration (delayed phase) at the pelvic level. Gas bubbles are recognizable in the lumen of both femoral veins (arrows).

**Figure 4 diagnostics-14-01425-f004:**
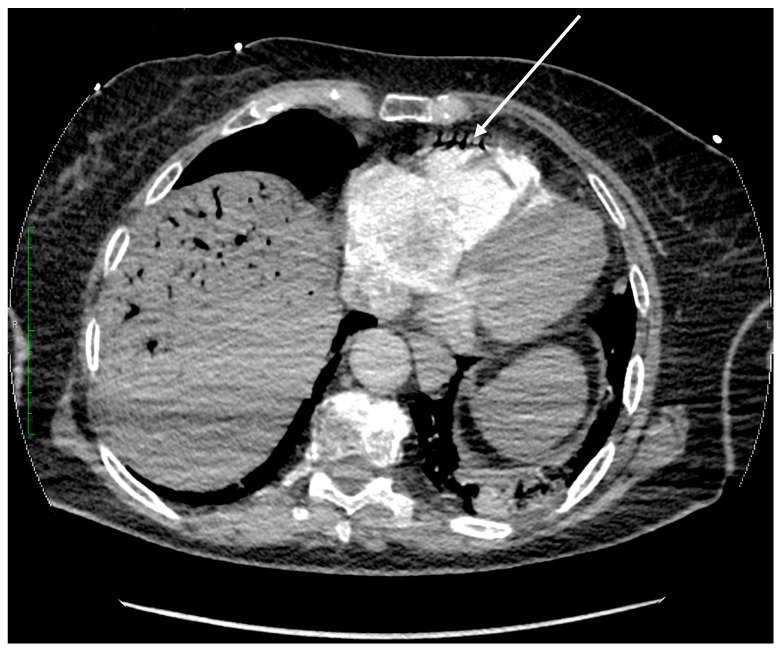
Axial plane of the chest CT angiography scan after intravenous contrast administration (soft tissue window). Massive aerobilia is observed within the biliary tree. Air is also present in the right ventricle (arrow).

**Figure 5 diagnostics-14-01425-f005:**
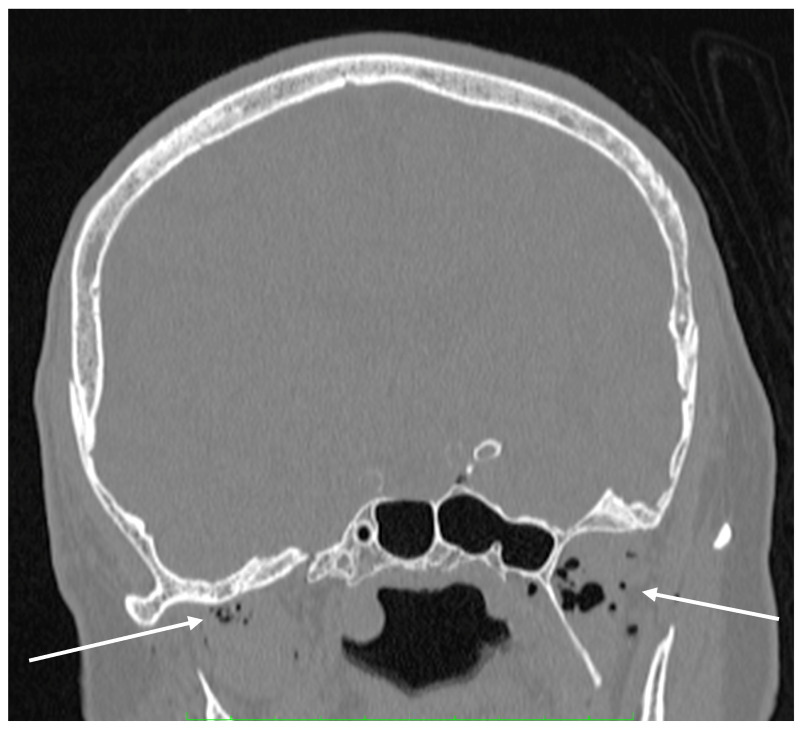
Coronal plane of the head’s CT scan. Gas bubbles are detected in the lumen of blood vessels of the face and intracranially (arrows).

**Table 1 diagnostics-14-01425-t001:** Laboratory results of a patient with gas embolism during ERCP. Day 1 was the day when the ERCP was performed and the patient was transferred to the Intensive Care Unit.

Day of Stay in the ICU /Parameter	Trombocytes 10^9^/L	Prokalcytonin ng/mL	CRP mg/mL
Day 1	116	0.0	104
Day 2	71	10.5	176
Day 3	59	4.8	77
Day 5	89	1.0	22
Day 7	147	0.5	11

## Data Availability

The datasets used and/or analyzed during the current study are available from the corresponding author upon reasonable request.

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
