# Peer review of "Analysis of Tomographic Images of a Catastrophic Gas Embolism during Endoscopic Retrograde Cholangiopancreatography"

_diagnostics, 2024, doi:10.3390/diagnostics14131425_

Round 1

Reviewer 1 Report

Comments and Suggestions for Authors

I read with great interest the manuscript by Frydrych et al., entitled "Analysis of Tomographic Images of a Catastrophic Gas 2 Embolism During ERCP".

Air embolism (a result of direct communication with the vasculature and an external pressure gradient from the gastrointestinal or the biliary tract), although rare, is a potentially devastating adverse event seen in ERCP procedures.

In this paper, the authors try to present the mechanism of gas bubble transmission into the vascular system through a series of angio-CT scan images: the presence of intramural gas in the duodenum and then to the portal venous system and to the IVC

Since air may be rapidly absorbed from the circulation while diagnostic tests are being arranged, imaging or invasive procedures are sometimes inaccurate. So CT imaging findings are not that common (as in this manuscript).

However, there are some major issues with the presentation.

Figures have no figure legends, no arrows or arrowheads making it nearly impossible for a non-diagnostician to follow. PLEASE CORRECT

The case as presented is very mixed up. First authors start with a single paragraph of the case and then they go back and forth to pathophysiology and their patient. IT WOULD BE MUCH EASIER FOR THE READER TO GO INTO PATHOPHYSIOLOGY ONCE THE CASE IS REPORTED

Table 1 has no legend. Also what about the days? instead of using dates, the authors should use day 1, 2, 3, etc. PLEASE REPHRASE.

Also they should point out the following CLEARLY (although more or less some of these points are discussed but in a chaotic manner. Air embolism can take the form of venous air embolism (where the embolus reaches the pulmonary circulation), systemic arterial air embolism (where the embolus reaches the coronary or the cerebral arteries), and paradoxical air embolism (where the venous embolus reaches the systemic circulation).

in cases of a venous air embolus the symptoms appear or get significantly worse upon repositioning the patient from the prone to the supine position after the procedure has ended

ABBREVIATIONS SHOULD BE EXPLAINED

Authors also state "The aim of this article is to draw attention to this complication" This cannot hold since air embolism during ERCP, although rare, is a well know complication to all endoscopists. PLEASE REPHRASE. The aim should to be the unique chance to see the imaging studies....

Author Response

Hello

Dear Reviewer, thank you very much for all comments

all necessary changes are added into the paper!

We believe that because fo your great expertise and some practice insights we improved the paper greately.

thank you

RESPONSES TO REVIEWS:

Thank you very much for taking the time to review this manuscript. We are very glad to hear that you read our article with interest. Please find the detailed responses below and the corresponding revisions in track changes in the re-submitted files.

Comment 1: Figures have no figure legends, no arrows or arrowheads making it nearly impossible for a non-diagnostician to follow.

Response 1: This is a very valid observation. In the revised manuscript each figure has a short legend indicating what it depicts. In addition, for ease of viewing the figures, arrows were used to show the presence of gas. However, I have tried to ensure that the arrows do not overshadow the rest of the diagnostic image, in order to simultaneously sensitize the reader to finding gas in the CT scan.

Comment 2: The case as presented is very mixed up. First authors start with a single paragraph of the case and then they go back and forth to pathophysiology and their patient. IT WOULD BE MUCH EASIER FOR THE READER TO GO INTO PATHOPHYSIOLOGY ONCE THE CASE IS REPORTED

Response 2: We have to admit that this is right. In the revised manuscript, we tried to organize the text more. To begin, we introduce the reader to the topic by reporting the case. Then due to the fact that this article was submitted to a special “Interesting Images” collection, we focused on presenting the figures with CT images and discussing etiology and pathophysiology. We paid attention to ensure that all the content had a cause-and-effect sequence.

Comment 3: Table 1 has no legend. Also what about the days? instead of using dates, the authors should use day 1, 2, 3, etc.

Response 3: Thank you for pointing out important details that are so easy to miss. Of course, I changed the content of the table from dates to days and added a legend.

Comment 4: Also they should point out the following CLEARLY (although more or less some of these points are discussed but in a chaotic manner. Air embolism can take the form of venous air embolism (where the embolus reaches the pulmonary circulation), systemic arterial air embolism (where the embolus reaches the coronary or the cerebral arteries), and paradoxical air embolism (where the venous embolus reaches the systemic circulation).

in cases of a venous air embolus the symptoms appear or get significantly worse upon repositioning the patient from the prone to the supine position after the procedure has ended

Response 4: This commentary definitely helped us to organize the text, so we decided to include the information contained in it in the improving the content. In the latest version of our manuscript, we tried to clearly guide the reader through the pathophysiology of embolism by referring to the described case and CT scans images.

Comment 5: ABBREVIATIONS SHOULD BE EXPLAINED

Response 5:  All abbreviations are explained on page 6 lines 224 – 233. By bringing this to my attention after revisiting the manuscript, I found three unexplained abbreviations and added.

Comment 6: Authors also state "The aim of this article is to draw attention to this complication" This cannot hold since air embolism during ERCP, although rare, is a well know complication to all endoscopists. PLEASE REPHRASE. The aim should to be the unique chance to see the imaging studies....

Response 6: This is a remarkably accurate perception. This is definitely the purpose of our article. The corresponding section of the text has been fixed.

We are very grateful for the opportunity to improve our article with this review.

Reviewer 2 Report

Comments and Suggestions for Authors

Dear authors, 

The case you presented is exceptional, but major revisioning paper reorganization are needed

English should be revised by a native person.

Introduction

You should define and present the pathology and its incidence.

Case presentation

You should introduce the patient.

Lab, previous examination and why he performed ERCP.

The contextualize the ERCP procedure and the adverse event.

Then the patient, once stable, underwent CT.

CT protocol is missing.

Figure legends are completely missing. Please reorder figures (it’s preferable not to jump from abdomen to thorax to brain and then pelvis). Consider to crop the black space around images and add arrow to point out the findings (that should be explained in the legends but also in the text)

After figure 5, I suppose your discussion starts.

The figure illustrates the onset of pathology caused by an ERCP complication: the presence of intramural CO2-gas in the duodenum. Subsequently, the gas is transmitted into the portal venous system through damaged duodenal vein radicles. 

The pathway of CO2-gas transmission also occurs by the biliary tract, the gas then 67 enters the vessels of the circulatory system through injured endothelium”

 This part is very interesting because you explain the air pathway, you should improve and underline these periods and remember to reorder and describe the figure following the soma logical reasoning,

Page 5 line 98 : the “CT” study?

Table 1 is not cited in the text and data are not preferred, you should “date” related to the event (hours and/or days after).

Summarize the discussion avoiding to repat the same concept with the exception of your teaching final message. 

Please summarize the importance of CT findings that reflects the air pathway.

Bibliography should be improved.

Please check this reference

Lanke G, Adler DG. Gas embolism during endoscopic retrograde cholangiopancreatography: diagnosis and management. Ann Gastroenterol. 2019 Mar-Apr;32(2):156-167. doi: 10.20524/aog.2018.0339. Epub 2018 Dec 20. PMID: 30837788; PMCID: PMC6394273.

It shows a nice diagram explained the air pathways, you may consider to make a similar diagram for your work that may significantly improve the paper.

Comments on the Quality of English Language

English revision needed.

Author Response

Hello

Dear Reviewer, thank you very much for all comments

all necessary changes are added into the paper!

We believe that because fo your great expertise and some practice insights we improved the paper greately.

thank you

RESPONSES TO REVIEWS:

Thank you very much for taking the time to review this manuscript. We are very glad that you find our described case exceptional. Please find the detailed responses below and the corresponding revisions in track changes in the re-submitted files.

Comment 1: English should be revised by a native person.

Response 1: Thank you that it has been brought to our attention. Our article has been rechecked and all errors found have been corrected in the latest version.

Comment 2: Introduction

You should define and present the pathology and its incidence.

Case presentation

You should introduce the patient.

Lab, previous examination and why he performed ERCP.

The contextualize the ERCP procedure and the adverse event.

Then the patient, once stable, underwent CT.

CT protocol is missing.

Response 2: Very fair observation that in the usual way of writing a case report, this should be in the introduction and case presentation. Due to the fact that this article was submitted to a special “Interesting Images” collection, we focused on presenting the figures with CT images and discussing etiology and pathophysiology. Which is why, at the beginning, we introduce the reader to the topic by reporting the case. We presented the most important information about our patient, including age, gender, cause of ERCP performed. In the revised manuscript, we tried to organize the text more. We paid attention to ensure that all the content had a cause-and-effect sequence.

Comment 3: Figure legends are completely missing. Please reorder figures (it’s preferable not to jump from abdomen to thorax to brain and then pelvis). Consider to crop the black space around images and add arrow to point out the findings (that should be explained in the legends but also in the text)

Response 3: This is a very valid observation. In the revised manuscript each figure has a short legend indicating what it depicts. In addition, for ease of viewing the figures, arrows were used to show the presence of gas. However, I have tried to ensure that the arrows do not overshadow the rest of the diagnostic image, in order to simultaneously sensitize the reader to finding gas in the CT scan. Following the good advice to improve visibility, the black space around the images has been cropped.

Comment 4: After figure 5, I suppose your discussion starts.

Response 4: In the latest version of our manuscript, we tried to clearly guide the reader through the pathophysiology of embolism by referring to the described case and CT scans images.

Comment 5: The figure illustrates the onset of pathology caused by an ERCP complication: the presence of intramural CO2-gas in the duodenum. Subsequently, the gas is transmitted into the portal venous system through damaged duodenal vein radicles. 

The pathway of CO2-gas transmission also occurs by the biliary tract, the gas then 67 enters the vessels of the circulatory system through injured endothelium”

 This part is very interesting because you explain the air pathway, you should improve and underline these periods and remember to reorder and describe the figure following the soma logical reasoning,

Response 5: Thank you for this insight, we hope that the revised version of the manuscript highlights the key parts of the air pathway. We have changed the order of the figures according to the order of the CT scans performer – abdomen and pelvis first, then the CT angiography of the chest and finally the CT of the head.

Comment 6: Page 5 line 98 : the “CT” study?

Response 6: Yes, there should be “CT”. Thank you for noticing,

Comment 7: Table 1 is not cited in the text and data are not preferred, you should “date” related to the event (hours and/or days after).

Response 7: Thank you for pointing out important details that are so easy to miss. Of course, I changed the content of the table from dates to days and added a legend. In the revised version of the manuscript, the table is cited in the text.

Comment 8: Summarize the discussion avoiding to repat the same concept with the exception of your teaching final message. Please summarize the importance of CT findings that reflects the air pathway.

Response 8: It is good that the repetitions were noticed, when developing the revised text we tried to avoid this to increase readability.

Comment 9: Bibliography should be improved.

Please check this reference

Lanke G, Adler DG. Gas embolism during endoscopic retrograde cholangiopancreatography: diagnosis and management. Ann Gastroenterol. 2019 Mar-Apr;32(2):156-167. doi: 10.20524/aog.2018.0339. Epub 2018 Dec 20. PMID: 30837788; PMCID: PMC6394273.

It shows a nice diagram explained the air pathways, you may consider to make a similar diagram for your work that may significantly improve the paper.

Response 9: The mentioned diagram indeed explains in detail the air pathways and proved to be very helpful in bringing order to the article. Due to the fact that this article was submitted to a special “Interesting Images” collection, we decided to focused on presenting the figures with CT images. We believe that there is no need to add another sixth figure, but we have included information on the air pathways in the revised version of the manuscript.

We are very grateful for the opportunity to improve our article with this review.

Round 2

Reviewer 1 Report

Comments and Suggestions for Authors

All comments have been addressed. 

Comments on the Quality of English Language

Minor editing would be appropriate

Author Response

Thank you very much for taking the time to once again review this manuscript. We are very glad that we were able to address all comments. Please find the detailed response below and the corresponding revisions in track changes in the re-submitted files.

Comment 1: Minor editing of English language required

Response 1: Thank you for bringing this to our attention. Our article has been rechecked and all identified errors have been corrected in the latest version by a colleague fluent in English writing.

We are very grateful for the opportunity to improve our article with this review.

Reviewer 2 Report

Comments and Suggestions for Authors

Dear Authors, 

Thank you for your improvement.

I suggest you just few more corrections, but this is because I really appreciate your work and I’d like you improve it in the best way it is possible. 

·       a CT scan of abdomen and pelvis 40 (Figures 1 – 3), an angio-CT scan of the chest (Figure 4) and a CT scan of head (Figure 5)”

As you described the CT examination it can appear she performed several CT studies. I suggest to change this period with

“Total Boby CT scan with intravenous contrast administration (Figure 1-5)”

·      One perplexity, the bowel is opacified. I’m sorry I did not notice it before. Did you administered oral contrast? It’s also in the stomach. You should explain this finding, 

·      Figure 1. Axial CT images (lung window) at the level of the liver. Extensive intrahepatic gas within the bile ducts is appreciable (arrow).

·       Figure 2. Axial CT images after intravenous contrast administration (delayed phase).  Air bubbles are detected within the duodenum wall (arrow).

·      Figure 3. Axial CT images at pelvis level (PLEASE CHECK THE CONTRAST PHASE).  Air bubbles are recognizable in the lumen of both femoral veins (arrows)

·      Figure 4. I would move this after figure 1. Axial CT image (PLEASE CHECK THE CONTRAST PHASE, in this image the right chambers are opacified, it should the angiographic chest phase. You can leave it or change with a similar phase that you used for the other image). Massive aereobilia (or penumobilia as you prefer) is appreciable within bile duct). Air is also appreciable in the right ventricle (arrow). 

·      Figure 5. Coronal CT image of the brain (bone window). Air bubbles are detected (arrows)(TRY TO SPECIFY THE LOCATION).

·      Please recall the figure in the discussion and remember to change the order of images (Figure 4 move up). It’s missing a figure of air embolism, but it’s ok.

Again, your case is exceptional. I hope to have the opportunity to cite your work soon.

Kind regards

Author Response

Thank you very much for taking the time to once again review this manuscript. We are very glad that you find our described case exceptional. We are excited by such a quick opportunity to have our work cited. Please find the detailed responses below and the corresponding revisions in track changes in the re-submitted files.

Comment 1: “a CT scan of abdomen and pelvis 40 (Figures 1 – 3), an angio-CT scan of the chest (Figure 4) and a CT scan of head (Figure 5)”

As you described the CT examination it can appear she performed several CT studies. I suggest to change this period with

“Total Boby CT scan with intravenous contrast administration (Figure 1-5)”

Response 1: This fix would definitely improve the reading of the text. In fact, several CT studies have been performed. Images from these CT scans were obtained separately, with separate descriptions. CT scans were carried out one by one, but nevertheless it was not Total Body CT scan. That is why we decided to keep the description in its current form.

Comment 2: One perplexity, the bowel is opacified. I’m sorry I did not notice it before. Did you administered oral contrast? It’s also in the stomach. You should explain this finding,

Response 2: Thank you very much for your perceptiveness. The contrast was administered only intravenously.

Comment 3:  Figure 1. Axial CT images (lung window) at the level of the liver. Extensive intrahepatic gas within the bile ducts is appreciable (arrow).

  • Figure 2. Axial CT images after intravenous contrast administration (delayed phase).  Air bubbles are detected within the duodenum wall (arrow).
  • Figure 3. Axial CT images at pelvis level (PLEASE CHECK THE CONTRAST PHASE).  Air bubbles are recognizable in the lumen of both femoral veins (arrows)
  • Figure 4. I would move this after figure 1. Axial CT image (PLEASE CHECK THE CONTRAST PHASE, in this image the right chambers are opacified, it should the angiographic chest phase. You can leave it or change with a similar phase that you used for the other image). Massive aereobilia (or penumobilia as you prefer) is appreciable within bile duct). Air is also appreciable in the right ventricle (arrow). 
  • Figure 5. Coronal CT image of the brain (bone window). Air bubbles are detected (arrows)(TRY TO SPECIFY THE LOCATION).

Response 3: We greatly appreciate such detailed information on how to improve the figure captions. We analyzed and corrected all of them.

Comment 4: Please recall the figure in the discussion and remember to change the order of images (Figure 4 move up). It’s missing a figure of air embolism, but it’s ok.

Response 4: Thank you for helping us better organize our writing. As I mentioned above, we presented the images according to how the separate CT studies were performed. Moving Figure 4, we will discuss abdominal CT, then CT angiography of the chest, and then jump to abdominal CT again. In the current arrangement, the first three figures are for abdominal CT, and then we move on to describe another study. We thought for a long time in what order to present the images from the CT scans, whether, for example, from the head to the limbs, but we chose this way according to the division of the CT studies performed.

We are very grateful for the opportunity to improve our article with this review.